# Application of Antipsychotic Drugs in Mood Disorders

**DOI:** 10.3390/brainsci13030414

**Published:** 2023-02-27

**Authors:** Janusz K. Rybakowski

**Affiliations:** Department of Adult Psychiatry, Poznan University of Medical Sciences, 60-572 Poznan, Poland; janusz.rybakowski@gmail.com

**Keywords:** antipsychotic drugs, first generation, second generation, third generation, bipolar mood disorder, mania, bipolar depression, maintenance treatment

## Abstract

Since their first application in psychiatry seventy years ago, antipsychotic drugs, besides schizophrenia, have been widely used in the treatment of mood disorders. Such an application of antipsychotics is the subject of this narrative review. Antipsychotic drugs can be arbitrarily classified into three generations. First-generation antipsychotics (FGAs), such as phenothiazines and haloperidol, were mainly applied for the treatment of acute mania, as well as psychotic depression when combined with antidepressants. The second-generation, so-called atypical antipsychotics (SGAs), such as clozapine, risperidone, olanzapine, and quetiapine, have antimanic activity and are also effective for the maintenance treatment of bipolar disorder. Additionally, quetiapine exerts therapeutic action in bipolar depression. Third-generation antipsychotics (TGAs) started with aripiprazole, a partial dopamine D2 receptor agonist, followed by brexpiprazole, lurasidone, cariprazine, and lumateperone. Out of these drugs, aripiprazole and cariprazine have antimanic activity, lurasidone, cariprazine, and lumateperone exert a significant antidepressant effect on bipolar depression, while there is evidence for the efficacy of aripiprazole and lurasidone in the prevention of recurrence in bipolar disorder. Therefore, successive generations of antipsychotic drugs present a diverse spectrum for application in mood disorders. Such a pharmacological overlap in the treatment of schizophrenia and bipolar illness stands in contrast to the dichotomous Kraepelinian division of schizophrenia and mood disorders.

## 1. Introduction

In 1952, French psychiatrists led by Jean Delay reported the therapeutic (antipsychotic) effect of the phenothiazine derivate, chlorpromazine. The date of this paper is widely accepted as marking the introduction of antipsychotic drugs to psychiatry [1]. Another important year in the history of these drugs is 1957, when the antipsychotic effect of haloperidol, a butyrophenone derivate, was described by Belgian psychiatrists [2]. Another significant date could be 1963, when Swedish pharmacologists suggested that the pharmacological mechanism of chlorpromazine and haloperidol is connected with their effect on the dopaminergic system of the brain [3].

After chlorpromazine, many aliphatic and piperazine derivatives of phenothiazine showing antipsychotic activity entered psychiatric treatment, including, e.g., fluphenazine, having the first long-acting injectable (LAI) preparation. Based on the structural similarity to phenothiazines, thioxanthene derivatives were also initiated as a group of antipsychotics. A few of the latter, namely, clopenthixol and flupentixol, have been among the most popular LAI antipsychotics in Europe [4]. Besides phenothiazine, thioxanthene, and butyrophenone substances, the antipsychotic drugs of benzamide structure should be mentioned, with the first of them being sulpiride, introduced in 1968 [5].

### 1.1. First-Generation Antipsychotic Drugs

The medications listed above are traditionally regarded as belonging to the first generation of antipsychotic drugs (FGAs). Their main mechanism of action, as suggested by Carlsson and Lindqvist for chlorpromazine and haloperidol, is connected with their effect on the dopaminergic system of the brain [3]. Nearly 60 years after this publication, it can be stated that such a mechanism has pertained to all antipsychotic drugs used to date. Some of the first-generation antipsychotics (e.g., haloperidol and sulpiride) can be regarded as relatively “pure” dopaminergic blockers, whereas many others also exert some effects on adrenergic, muscarinergic, and histaminergic receptors.

The dopaminergic concept of antipsychotic action has undergone, in subsequent years, a gradual refinement. In 1966, van Rossum put forward a hypothesis that the mechanism of antipsychotic drugs is due to the blockade of dopaminergic receptors [6]. A decade later, Seeman et al. [7] demonstrated that the main dopamine receptors involved in antipsychotic activity are D2 ones. In the last decade, support for a connection between the dopaminergic D2 receptor and psychosis (schizophrenia) was also provided from a major molecular genetic study, where the D2 receptor gene was one of the 108 schizophrenia-associated genetic loci [8].

### 1.2. Second-Generation Antipsychotic Drugs

The so-called second-generation or atypical antipsychotics (SGAs), with the exception of clozapine, were mostly introduced in the 1990s. The pharmacological difference between second- and first-generation antipsychotics could be that the former, besides the dopaminergic blockade, also exerts a significant effect on other neurotransmitter systems, mainly serotonergic (5-HT). On the clinical side, second-generation antipsychotic drugs are supposed to induce fewer motor side effects and have a broader spectrum of therapeutic activity in schizophrenia, targeting, e.g., cognitive and negative symptoms. The latter symptoms are usually improved with low doses of these drugs, while higher doses are most efficient in treating positive (psychotic) symptoms. Another difference with the first generation of antipsychotics may be that the second-generation ones exert various degrees of neuroprotective effects, often in a dose-dependent manner [9].

It transpired that the real precursor of atypical antipsychotic drugs was clozapine. The drug was introduced in Europe in the 1970s, although preclinical studies did not indicate its “neuroleptic” activity. On the other hand, the observations of the clinical action of clozapine contradicted the belief of the time that an antipsychotic effect should be accompanied by extrapyramidal symptoms [10]. In 1975, shortly after the introduction of the drug in Finland, there was an epidemic of 16 cases of agranulocytosis in patients treated with clozapine. Among them, eight subjects died due to a secondary infection [11]. Following this event, in the late 1970s, clozapine was suspended for several years, and after re-introduction, the requirement of frequent monitoring of the leukocyte system was established. At the turn of 1989/1990, the use of clozapine was initiated in the USA, where it experienced a therapeutic revival. It was found that the drug was very efficient in treatment-resistant schizophrenia [12]. Secondly, the main pharmacological mechanism of clozapine was suggested as involving the blockade of both D2 and 5-HT2 receptors, paving the way to such an interpretation of mechanisms for further atypical (second-generation) antipsychotics [13]. Thirdly, the observations in mood disorders, mainly in bipolar mania, resulted in a suggestion that the drug may possess mood-stabilizing properties [14]. Thus, clozapine became the first representative of the so-called second generation of mood-stabilizing drugs and preceded other atypical antipsychotics in this respect.

Second-generation antipsychotic drugs, besides clozapine, include amisulpride, asenapine, olanzapine, paliperidone, quetiapine, risperidone, sertindole, and ziprasidone. The family of these diverse drugs shows significant intraindividual differences concerning affinity to various receptors and their influence on motor and metabolic symptoms, as well as their effect on prolactin levels. As far as receptor affinity is concerned, an exception could be amisulpride, which does not exert any effect on serotonergic receptors. Other representatives of this group all act on serotonergic 5-HT2 receptors but show differences in their effect on other serotonin receptors, as well as histamine, adrenergic, and muscarinic ones. Recently, Aringhieri et al. [15] suggested a spectrum of atypical drugs as a continuum, with clozapine on the one side (the most atypical) and risperidone on the other side (the least atypical). 

For the aim of this review, it is important to note that besides clozapine, several SGA fulfilled the criteria for “mood stabilizer”. I propose a definition of a mood stabilizer that considers the role of a drug in the acute and long-term treatment of bipolar disorder. The first aspect of such a definition should contain a reduction or amelioration in manic and/or depressive symptoms during an acute episode. The second aspect requires the prevention of manic and/or depressive recurrences during long-term administration when a drug is given as monotherapy, and a trial is performed for at least 1 year. The third important aspect is that the drug should not induce or worsen either manic or depressive episodes or mixed states [16]. Having such a definition in mind, I suggested, in 2010, a classification of mood stabilizers based on the chronology of their introduction into the psychiatric armamentarium. The first generation of mood stabilizers, such as lithium, valproates, and carbamazepine, came in the 1960s and 1970s. It was not until 1995 that the mood-stabilizing property of clozapine was discovered [14], which started the second generation of mood stabilizers [17]. The criteria for mood stabilizers, according to the definition above, were fulfilled by olanzapine and quetiapine in the second half of the 1990s and by risperidone in 2010 [18]. However, in this review, the therapeutic effects of potential mood stabilizers will be discussed separately for acute episodes and maintenance treatment.

### 1.3. Third-Generation Antipsychotic Drugs

It can be assumed that the concept of the third generation of antipsychotics (TGAs) started with the introduction of aripiprazole because the drug was distinct from all previous antipsychotics with respect to the effect on dopamine neurotransmission, being the partial agonist of dopamine D2 receptors [19]. However, other features of the TGA were indicated, such as the effects on dopamine D3 receptors and various serotonin receptors, especially 5-HT1A and 5-HT7 [20]. On the clinical side, the pro-cognitive and anti-negative effects of TGA in schizophrenia were found to be positively associated with the dose, while in the case of SGA, they showed a negative correlation.

To be more specific, both aripiprazole and brexpiprazole are partial agonists of D2, D3, and 5-HT1A receptors and antagonists of 5-HT2A, 5-HT2C, and 5-HT-7 receptors. The difference is that brexpiprazole is also an antagonist of alpha-1 adrenergic and H1 histaminergic receptors, which makes it a little more sedative than aripiprazole. The next partial agonist of D2, D3, and 5-HT1A receptors is cariprazine, which is also an antagonist of 5-HT2B receptors. Cariprazine has also the highest affinity to D3 receptors [21]. Lurasidone and lumateperone are also counted as belonging to the TGAs although both are antagonists of D2 receptors. However, besides this, lurasidone is an antagonist of 5-HT2A and 5-HT7 receptors and a partial agonist of 5-HT1A receptors, and lumateperone is a partial agonist of D3 and 5-HT1A receptors and an inhibitor of serotonin transporters [22].

Since their introduction into the psychiatric armamentarium seventy years ago, antipsychotic drugs have been widely used therapeutically in various aspects of mood disorders. In this narrative review, such uses will be reviewed and discussed. The antipsychotic drugs will be arbitrarily classified into three generations, as suggested above. After presenting the clinical applications of antipsychotic drugs in mood disorders, the underlying pharmacological mechanisms will be discussed. Finally, the evolution of antipsychotic drugs will be placed in the context of a possible overlap between schizophrenia and bipolar disorder. 

## 2. Application of the First Generation of Antipsychotic Drugs

### 2.1. Mania

As far as mood disorders are concerned, the "classical " antipsychotic drugs, after their launch into psychiatry, found their leading application in the treatment of mania. During this time, the therapeutic effect of lithium on mania became known. In 1949, John Cade, an Australian psychiatrist, described a favorable effect of lithium carbonate in ten manic patients [23]. The antimanic effect of lithium was further demonstrated in a larger group of patients by Australian investigators [24] and in a placebo-controlled study by Danish researchers led by Mogens Schou in 1954 [25]. The usefulness of chlorpromazine, as well as indications for the employment of this drug in mania, was voiced shortly after its initiation in psychiatry, even before the introduction of haloperidol [26,27]. About two decades later, a comparison of chlorpromazine and lithium carbonate performed by Japanese researchers found this antipsychotic drug inferior to lithium in its antimanic efficacy [28]. Additionally, a comparison of the efficacy of chlorpromazine with electroconvulsive therapy (ECT) in mania showed that, although both treatment modalities were effective, ECT was also helpful in chlorpromazine nonresponders [29].

Haloperidol, the other most important antipsychotic drug of the first generation, in the years after its initiation in psychiatry, showed high efficacy in the treatment of mania. The effect was especially rapid when the drug was given intramuscularly or intravenously [30]. In a comparative study performed in 1975, haloperidol was similar to lithium in causing a highly significant improvement in manic symptoms without sedation [31]. More than three decades later, in a meta-analysis of the comparative efficacy of antimanic drugs, haloperidol outperformed both typical and atypical antipsychotic drugs, as well as lithium and anticonvulsants [32]. From the current perspective, haloperidol has remained one of the most valuable and effective drugs for the treatment of mania [33]. The possibility of using haloperidol in intramuscular injections and LAI form additionally emphasizes its usefulness. 

### 2.2. Psychotic Depression

As another indication for the use of the first generation of antipsychotic drugs, the treatment of delusional depression, both unipolar and bipolar, emerged. In such situations, a combination of typical antipsychotics with tricyclic antidepressant drugs was attempted. The most frequent composite was that of antipsychotic perphenazine and antidepressant amitriptyline [34]. However, favorable experiences with adding haloperidol as well as zotepine to an antidepressant were also reported [35].

## 3. Application of Second-Generation Antipsychotic Drugs

### 3.1. Mania

#### 3.1.1. Clozapine

Clozapine is a highly effective drug in the treatment of acute manic episodes. We observed the antimanic activity of clozapine more than four decades ago [36]. However, no randomized controlled trials have been performed with clozapine in mania, and its use is “off-label”. Nevertheless, a recent systematic review and meta-analysis showed that the antimanic efficacy of clozapine is similar to other antipsychotics and superior among treatment-resistant cases [37]. The attempt of using clozapine in treatment-resistant mania was also advised by leading experts in bipolar disorder [33,38]. The latest Polish standards of pharmacological treatment of mania suggest the use of clozapine in mania after a lack of significant improvement following 4–8 weeks of treatment with other antipsychotics or/and mood stabilizers [39].

#### 3.1.2. Olanzapine 

In the treatment of mania, excellent clinical efficacy and good tolerance of olanzapine, in doses of 10-20 mg/day, have been demonstrated [40]. According to a comparative analysis, olanzapine exhibits the best profile of efficacy and tolerability [32]. In manic episodes, short-acting injections of the drug are very useful. Olanzapine also obtained Food and Drug Administration (FDA) approval for the treatment of manic/mixed episodes in pediatric bipolar disorder [41]. Olanzapine, combined with lithium or valproate, potentiates their therapeutic effects on mania concerning psychotic, manic, mixed, and depressive symptoms [42].

#### 3.1.3. Quetiapine

In the manic state, quetiapine is effective at doses of 400-800 mg/day, and its efficacy is similar to that of lithium. Besides adult patients, quetiapine also obtained FDA approval for manic/mixed episodes in pediatric bipolar subjects [41]. A combination of quetiapine with lithium or valproate is significantly more effective in the treatment of manic episodes than any of these first-generation mood stabilizers used in monotherapy [43].

#### 3.1.4. Risperidone 

Like other atypical antipsychotics, risperidone at doses of 1-6 (average 3-4) mg/day shows significant antimanic properties [44]. Risperidone, second to olanzapine, shows the best efficacy and tolerance in manic episodes [32]. The drug is also approved by FDA for manic/mixed episodes in pediatric bipolar illness [41]. A combination of risperidone with lithium or valproate is significantly more effective in the treatment of manic episodes than any of these mood stabilizers alone [45].

#### 3.1.5. Asenapine

Asenapine, at doses of 10-20 mg/day shows good efficacy and tolerability in the treatment of mania, as was confirmed in a meta-analysis published in 2013 [46]. In the recent guidelines of the Canadian Network for Mood and Anxiety Treatments (CANMATs) created by a panel of the most eminent international experts, asenapine is considered one of the most important drugs for the treatment of manic episodes [47]. The drug is approved both in Europe and in the USA for the treatment of manic or mixed episodes of adult and pediatric bipolar illness [48]. Asenapine when added to mood stabilizer monotherapy significantly augments the effect of treatment [49].

#### 3.1.6. Ziprasidone

Ziprasidone was also found efficacious and had good tolerability in mania [50]. For psychomotor agitation in mania, intramuscular injections of the drug could be also useful. On the other hand, out of all atypical antipsychotics, the use of ziprasidone was most probably associated with the induction of manic symptoms [51].

### 3.2. Prophylaxis of Bipolar Disorder

#### 3.2.1. Clozapine

Similar to mania, no randomized controlled trials have been performed with the long-term use of clozapine, and its employment in such indication is “off-label”. However, clinical experience indicates the usefulness of clozapine in the prevention of severe and drug-resistant forms of bipolar affective disorder. A systematic review of long-term clozapine use in the treatment of drug-resistant bipolar affective disorder covering 15 studies with over 1000 patients was performed in 2015. Clozapine monotherapy or its combination with other mood-stabilizing drugs resulted in a significant improvement in terms of manic symptoms, depression, psychosis, and rapid cycling, and a significant proportion of the patients experienced a significant general improvement or even remission [52]. The latest Polish standards suggest the long-term use of clozapine in treatment-resistant bipolar patients, with adequate monitoring of the hematological system. Concomitant use of lithium can diminish the risk of leukopenia. The predictive factor for good prophylactic efficacy of clozapine in bipolar affective disorder is severe manic episodes with psychotic symptoms and significant agitation. [39].

#### 3.2.2. Olanzapine 

Many controlled studies have been performed on the prophylactic efficacy of olanzapine monotherapy in bipolar affective disorder in comparison to placebo and first-generation mood-stabilizing drugs. The prophylactic efficacy of olanzapine monotherapy is high, mainly concerning manic rather than depressive episodes. It is particularly true in patients for whom olanzapine has been effective during an acute manic episode [53]. In a comparative study on olanzapine and lithium, the rate of mania recurrence was significantly lower for olanzapine [54]. In Polish standards, olanzapine monotherapy is considered to be the first-line treatment for long-term therapy in type I bipolar disorder with a predominance of manic episodes [39]. The efficacy of therapy combining olanzapine with lithium or valproate has also been studied showing that the prophylactic effect was augmented after adding olanzapine to lithium or valproate monotherapy [55]. The application of LAI olanzapine could further increase its prophylactic effectiveness, although no controlled trial on this issue has been performed yet. Additionally, because the main adverse somatic result of olanzapine is weight gain, and mood disorder patients are more prone to this effect, a combination of olanzapine with samidorphan may be helpful for many patients [56].

#### 3.2.3. Quetiapine 

Unlike other antipsychotics of the second generation, preventing mostly manic recurrences, quetiapine monotherapy in bipolar mood disorder effectively prevents both manic and depressive episodes. In the prevention of depression recurrences, quetiapine monotherapy was similarly efficacious as lithium monotherapy [43]. A better prophylactic effect after adding quetiapine to lithium or valproate monotherapy has also been found [57]. A four-year comparison of the prophylactic efficacy of quetiapine in monotherapy or combination with other mood stabilizers showed that the percentage of patients without recurrences on quetiapine monotherapy was 29.3%. However, when combining quetiapine with lithium, no recurrences occurred in 80% and, with valproate, in 78.3% of patients [58]. 

#### 3.2.4. Risperidone 

Beneficial long-term effects of risperidone on bipolar affective disorder were found when the drug was added to first-generation mood-stabilizing drugs [59]. Additionally, in 2010, a study lasting 2 years indicated that risperidone monotherapy in the form of LAI showed a significant prophylactic effect concerning recurrences of manic episodes in patients with type I bipolar affective disorder [60].

#### 3.2.5. Paliperidone

In a randomized, long-term placebo-controlled study, paliperidone extended-release tablets significantly prevented manic but not depressive episodes in patients with bipolar I disorder when started after an acute manic or mixed episode [60].

#### 3.2.6. Asenapine

The long-term prophylactic efficacy of asenapine was investigated in a randomized, placebo-controlled study. The drug significantly prevented both manic and depressive episodes in patients with bipolar I disorder when started after an acute manic or mixed episode [61]. Additionally, the addition of asenapine to lithium or valproate significantly augmented their long-term effect [49].

#### 3.2.7. Ziprasidone

There is no study on ziprasidone monotherapy for the maintenance treatment of bipolar disorder. The addition of ziprasidone to monotherapy with lithium or valproate improved their efficacy during 6-month follow-up [62].

### 3.3. Depression

#### 3.3.1. Quetiapine 

In placebo-controlled studies, quetiapine at a dose of 300 or 600 mg/day was found to be effective in the treatment of depression in the course of bipolar affective disorder type I or type II. Comparative studies have shown that the efficacy of quetiapine in this case may be greater than that of lithium [63]. In most guidelines, quetiapine has been recommended as a monotherapy for the treatment of bipolar depression [47]. The drug can also augment the efficacy of antidepressant drugs in treatment-resistant depression both bipolar and unipolar [64]. 

#### 3.3.2. Olanzapine 

In bipolar depression, olanzapine exerts a significant therapeutic effect when combined with fluoxetine. The efficacy of composite olanzapine–fluoxetine treatment of drug-resistant depression and psychotic depression in the course of both bipolar and unipolar affective disorder has also been demonstrated [65]. Olanzapine can also serve for the augmentation of antidepressant drugs in treatment-resistant depression [64].

#### 3.3.3. Risperidone

Risperidone does not exert an antidepressant effect. However, low doses of the drug can be used for the augmentation of antidepressants in treatment-resistant depression [64].

#### 3.3.4. Asenapine

In a post hoc analysis, it was found that asenapine reduced depressive symptoms in bipolar I patients treated for manic or mixed episodes [66]. Recently, a therapeutic effect of asenapine was reported in a small number of patients with bipolar depression [67] 

## 4. Application of Third-Generation Antipsychotics Drugs

### 4.1. Mania

#### 4.1.1. Aripiprazole

Aripiprazole at doses of 15-30 mg/day exerts a significant antimanic effect, and its therapeutic efficacy is similar to that of lithium. In manic episodes, short-acting injections of this drug can be also useful. A combination of aripiprazole with lithium or valproate is significantly more effective than any of these first-generation mood-stabilizing drugs used in monotherapy [68].

#### 4.1.2. Cariprazine 

In a double-blind, placebo-controlled study, cariprazine in both low (3-6 mg/day) and high (6-12 mg/day) doses were efficacious in the treatment of acute and mixed mania [69]. In a recent study, the effectiveness of carbamazepine in combination with lithium or valproate in first-episode mania was also demonstrated [70].

### 4.2. Depression

#### 4.2.1. Lurasidone

Many studies have demonstrated that, in bipolar depression, treatment with lurasidone monotherapy adjunctive to lithium or valproic acid, in doses of 20 to 120 mg once daily, results in a statistically and clinically significant reduction in depressive symptoms [71]. This was confirmed in meta-analyses and systematic reviews [72].

#### 4.2.2. Cariprazine

The FDA approved cariprazine for the treatment of bipolar depression in 2019 based on the results of three randomized, double-blind, placebo-controlled trials. The results suggested that the drug is an effective and well-tolerated treatment for bipolar depression when used in doses of 1.5-3 mg/day [73].

#### 4.2.3. Lumateperone

Lumateperone at 42 mg/day was recently studied in a phase 3 randomized, placebo-controlled trial. It was found that the drug significantly improved depression symptoms and was generally well tolerated in patients with major depressive episodes associated with both bipolar I and bipolar II disorders [74].

#### 4.2.4. Aripiprazole 

Aripiprazole in smaller doses (5-10 mg/day) can be used for the augmentation of antidepressants in case of their sub-optimal efficacy. In the United States, the drug has been recommended for the augmentation of antidepressants in drug-resistant depression [75].

### 4.3. Prophylaxis of Bipolar Disorder

#### 4.3.1. Aripiprazole 

A prophylactic effect of aripiprazole monotherapy was demonstrated in a two-year study, where the drug significantly prevented the recurrence of manic episodes but not depressive ones [76]. A significantly better prophylactic effect after adding aripiprazole to lithium or valproate monotherapy was also obtained [77].

#### 4.3.2. Lurasidone

A large study with up to two years of lurasidone administration in a flexible dose of 20-120 mg, following its use in bipolar depression, showed that the drug was safe and prophylactically efficacious when employed as monotherapy or as an adjunct to lithium or valproate [78]. Lurasidone (20-80 mg/day) combined with lithium or valproate showed a better prophylactic effect on bipolar I disorder than lurasidone placebo [79]. In a most recent 52-week study of Japanese patients with bipolar disorder, lurasidone at 20-120 mg/day, with or without lithium or valproate, maintained improvements in depressive symptoms for previously treated bipolar depressed patients and led to improvements in manic symptoms among a newly recruited subgroup of patients with a recent/current manic, hypomanic, or mixed episode [80].

## 5. Comparison of Second- and Third-Generation Antipsychotics in Mood Disorders

### 5.1. Comparison of the Efficacy in Mania

A recent meta-analysis of randomized controlled trials in bipolar mania, including some SGAs and TGAs, showed that aripiprazole, asenapine, cariprazine, olanzapine, paliperidone, quetiapine, risperidone, and ziprasidone were significantly better than placebo in terms of the response. A low all-cause discontinuation rate, reflecting good tolerance, was characteristic of aripiprazole, olanzapine, quetiapine, and risperidone [81]. 

### 5.2. Comparison of the Efficacy in Bipolar Depression

A meta-analysis of randomized controlled trials in bipolar depression, including SGAs and TGAs, showed that cariprazine, lurasidone, olanzapine, olanzapine–fluoxetine, and quetiapine were significantly better than placebo when used in various types and periods of bipolar illness [82].

### 5.3. Comparison of the Prophylactic Efficacy in Bipolar Disorder

Kishi et al. [83] performed meta-analyses of the prophylactic efficacy in bipolar disorder, including SGAs and TGAs. They found that aripiprazole, asenapine, olanzapine, paliperidone, and risperidone long-acting injections outperformed placebo in preventing the recurrence of any mood episode and also manic and depressive ones. An analysis of antipsychotics combined with lithium or valproate (LIT/VAL) showed that all antipsychotics other than olanzapine outperformed LIT/VAL placebo for preventing any mood episode. Lurasidone plus LIT/VAL and quetiapine plus LIT/VAL outperformed the LIT/VAL placebo for preventing depressive episodes, while aripiprazole plus LIT/VAL and quetiapine plus LIT/VAL were better than the LIT/VAL placebo for preventing mania. Lurasidone plus LIT/VAL and quetiapine plus LIT/VAL outperformed LIT/VAL placebo for all-cause discontinuation.

### 5.4. Use of LAI SGAs and TGAs for Maintenance Treatment of Bipolar Disorder

In recent years, the use of LAI antipsychotics has become more and more frequent in the long-term management of bipolar disorder. In a systematic review, LAI antipsychotics (risperidone and aripiprazole) were found to be well tolerated and effective for the treatment of manic symptoms and for preventing mood recurrences in adult patients with bipolar disorder [84]. In a recent mirror-image study including various LAI antipsychotics in adults with bipolar disorder, Bartoli et al. [85] found a significant decrease in terms of 12-month hospitalization rates and the number of days. 

## 6. Mechanism of Therapeutic Action of Antipsychotic Drugs in Mood Disorders

The mechanism of the therapeutic action of antipsychotic drugs in mania and depression can be explained based on a dopaminergic concept of bipolar disorder, suggesting an increase in dopaminergic activity in mania and a decrease in depression. Such a catecholaminergic hypothesis was proposed by Schildkraut in 1966 [86] and updated by Bunney in 1976 [87]. In 2017, the dopaminergic concept of bipolar disorder was examined by Ashok et al. [88]. They considered striatal dopaminergic D2/D3 receptor availability, explaining elevated dopaminergic neurotransmission in mania, and an increase in striatal dopamine transporter (DAT), elucidating lowered dopaminergic function in depression. The latter finding was corroborated in a recent study by Yatham et al. [89], showing reduced DAT in the right putamen and nucleus accumbens during an acute manic episode, suggesting dopaminergic overactivity. Therefore, the therapeutic action in mania, as well as the prevention of manic episodes in bipolar disorder, can be connected with the anti-dopaminergic activity, mainly by blocking D2 receptors, which manifests in all three generations of antipsychotic drugs.

Many studies have suggested that antidepressant activity may be connected with the inhibition of noradrenaline, dopamine, and/or serotonin transporters and, additionally, with a specific effect on some serotonergic receptors. The antidepressant effect of quetiapine may be associated with blocking the noradrenaline transporter and the activation of serotonin receptor 5-HT1A by its main active metabolite—norquetiapine—as is the case with many antidepressants [90]. Such an effect of lurasidone could be related to the activation of the serotonin receptor 5-HT1A and the antagonism of 5-HT2A and 5-HT7 receptors. As far as cariprazine is concerned, such an effect may be connected with the partial agonism of D2, D3, and 5-HT1A receptors. Finally, lumateperone can exert this effect by the partial agonism of D3 and 5-HT1A receptors and the inhibition of the serotonin transporter similar to antidepressants—selective serotonin reuptake inhibitors (SSRIs). Such antidepressant effects of the drugs in an acute episode may also act to prevent depressive episodes during maintenance treatment. 

## 7. Pharmacological Overlap between Schizophrenia and Bipolar Disorder

The German psychiatrist, Emil Kraepelin (1856–1926), in 1899, proposed the dichotomous separation of psychiatric disorders, which became a diagnostic basis for the 20^th^ century. He defined two classes of illnesses. The first, characterized by the chronic course and systematic cognitive decline, was named *“dementia praecox”*. The second, running periodically, without a distinct cognitive deterioration was termed *“Manisch-depressives Irresein”* [91]. After several years, *“dementia praecox”* was conceptualized as "schizophrenia" by the Swiss psychiatrist, Eugen Bleuler [92]. *“Manisch-depressives Irresein"* can be currently regarded as similar to "bipolar mood disorder". In an article published in 2019, the author of this review discussed this dichotomy while taking into account the latest findings, including psychopharmacological ones [93].

From the pharmacological point of view, the distinct therapeutic effect on schizophrenia and mood disorders of first-generation antipsychotics, antidepressants, as well as the first generation of mood stabilizers in schizophrenia and mood disorders, could support a dichotomous distinction between these illnesses. The therapeutic effect of antipsychotic drugs on both schizophrenia and mania could be explained based on altered dopaminergic transmission in both these conditions. However, the introduction of the SGAs and TGAs into psychiatry disrupted this dichotomy. These drugs were initially supposed to have their main application in schizophrenia; however, it turned out that a majority of them possess mood-stabilizing properties such as the prevention of bipolar mood episodes. Furthermore, some of them, such as quetiapine, lurasidone, cariprazine, and lumateperone, have been demonstrated to exert antidepressant activity in bipolar depression. Therefore, the specific pharmacological profile and clinical activity of the SGAs and TGAs places them somewhere in between the therapeutic tools for schizophrenia and bipolar disorder.

## 8. Conclusions

Antipsychotic drugs have been used in the treatment of mood disorders since their introduction seventy years ago. Their first generation was mainly applied for the treatment of acute mania, as well as psychotic depression when combined with antidepressants. However, the development of antipsychotic drugs into the second and third generations during the last three decades saw many of them being increasingly used for the treatment of both acute episodes and long-term prophylaxis of bipolar disorder. Antipsychotics of the second generation besides antimanic activity are also effective for the maintenance treatment of bipolar disorder, and quetiapine has a therapeutic effect on bipolar depression. Out of the third-generation antipsychotics, aripiprazole and cariprazine have antimanic activity, lurasidone, cariprazine, and lumateperone exert a significant antidepressant effect on bipolar depression, while aripiprazole and lurasidone have displayed evidence for the prevention of recurrence in bipolar disorder. It can be concluded that the successive generations of antipsychotic drugs present a diverse spectrum for application in mood disorders

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
