# Peer review of "Application of Antipsychotic Drugs in Mood Disorders"

_brainsci, 2023, doi:10.3390/brainsci13030414_

Round 1

Reviewer 1 Report (Previous Reviewer 2)

No further comments

Author Response

Thank you.

Reviewer 2 Report (Previous Reviewer 1)

1.     The meaning of the abbreviations “SGA” and „TGA” is not mentioned in the main text. Please define upon its first mention.

2.     Line 190: Not only could the use of clozapine in bipolar disorder be considered “off-label”, it literally IS off-label in most countries. Please change this statement.

3.     Lines 216–217: In which countries is risperidone in-label for this indication?

Author Response

  1. The meaning of the abbreviations “SGA” and „TGA” is not mentioned in the main text. Please define upon its first mention.

Response: This was done in lines 65 and 123.

  1. Line 190: Not only could the use of clozapine in bipolar disorder be considered “off-label”, it literally IS off-label in most countries. Please change this statement.

Response: The statement was changed in lines 189 and 236.

  1. Lines 216–217: In which countries is risperidone in-label for this indication?

    Response: The drug was approved by FDA for manic/mixed episodes in pediatric bipolar illness (line 214).

This manuscript is a resubmission of an earlier submission. The following is a list of the peer review reports and author responses from that submission.

Round 1

Reviewer 1 Report

General comments:

-          Overall an interesting an informative manuscript that provides a compact overview of the evidence of antipsychotic drug use in various states of affective disorders. It is very well-structured and this easy to follow. I also enjoyed the historical aspects the author mentions.

-          The language in this manuscript would benefit from editing by a native speaker.

-          I appreciate the receptor pharmacological approach the author has taken and I personally believe that it is invaluable to understand how drugs work on the different receptor/receptor systems. However, I assume many readers are not as familiar with the receptors and their functions. Perhaps a brief table explaining their therapeutic effects when agonists or antagonists are used would be helpful.

-          The author of this manuscript has dedicated much work into the analysis of clozapine’s use in bipolar disorder. While I have heard that clozapine (like all other second generation antipsychotic drugs, SGA) has mood-stabilizing properties, I am not aware that this approach is considered routine clinical practice. It is also “off-label” as the summary of product characteristics reserves the use of clozapine for treatment-resistant schizophrenia and schizoaffective disorders. This should be mentioned as it can have serious legal implications!!

-          Several times I read “atypical” or “typical” antipsychotic drugs vs first-/second-generation. Please use these terms homogenously. Also consider introducing an abbreviation such as SGA for second-generation and FGA für first generation. It would improve readability.

1. Introduction:

-          I do not understand what the first sentence means. Please rephrase to make it more comprehensible.

-          Please mind the use of correct tenses in the first paragraph: “As the date of the introduction of antipsychotic drugs to psychiatry, the publication of French psychiatrists led by Jean Delay in 1952 is assumed, showing the therapeutic (antipsychotic) effect of phenothiazine derivate, chlorpromazine [1]. An important year in the history of these drugs would be is also 1957 when the antipsychotic effect of haloperidol, the butyrophenone derivate, was demonstrated by Belgian psychiatrists [2]. Another significant date could be is 1963 when the publication of Swedish pharmacolo gists appeared suggesting that the pharmacological mechanism of chlorpromazine and 33 haloperidol is connected with their effect on the dopaminergic system of the brain [3].”

1.2. Second-generation of antipsychotic drugs

- I recommend starting this section with the development of the first SGA, i.e. clozapine

- “where it boomed a second childhood” is very odd phrasing

- Line 90: Clozapine does not have a particularly high affinity for D2-receptors

- Line 95: Clozapine is definitely not a mood-stabilizing drug and its use as such is – at least currently– most certainly off-label! This sentence should be more clearly phrased, so it becomes more apparent, that clozapine was the first SGA that was found to have “mood-stabilizing” properties.

- Line 95: perhaps better to say “chemically unrelated”

- Lines 108–: what are the criteria for “mood stabilizers”?

- Lines 120–122: poor language, please rephrase

1.3. Third generation of antipsychotic drugs

- Lines 124–132: please provide appropriate references

- Lines 134: instead of “are partial agonists of X” “act as partial agonists on XY”

- Lines 137–139: it would also be of value to mention that cariprazine has the highest affinity to D3-receptors

- Lines 144: “Since their entered” is not correct English; perhaps you mean to say “since their introduction”?

2.1. Mania

- Lines 154–167: very interesting historical aspects!

- Lines 169: Instead of calling haloperidol the “other most important”antipsychotic drug, it would be more objective to write that it was a “clinically valuable” antipsychotic drug.

- line 175: Which other “typical” and “atypical” antipsychotics were considered in the mentioned meta-analysis?

- line 176–178: At least from where I am from, haloperidol is rarely used in the treatment of mania – this is not to say it isn’t effective. If haloperidol really does find significant use in mania today, please provide an appropriate reference.

2.2. psychotic depression

- lines 180–181: “turned out” is informal English, please rephrase

- line 183: “were used” implies that this was standard procedure in the past. When was this the case?

3.1.1. Clozapine

- line 193–195: I would not go as far as to state that clozapine is “generally recommended” for the treatment of treatment-refractory mania, as this recommendation (to my knowledge) does not appear in the official treatment guidelines of other countries (NICE, etc.). Further, clozapine is solely in-label for treatment-resistant schizophrenia. In so, the use in bipolar patients is absolutely off-label, even though it may prove effective for this indication.

3.2.1. Clozapine

- lines 244–245: In most countries/according to the guidelines I am aware of, clozapine is not considered “second-line”treatment. The CANMAT guidelines you cited earlier consider clozapine “third-line” and clozapine has the lowest level of evidence for maintenance treatment of bipolar disorder. Further, clozapine’s risk for agranulocytosis is hardly the only concerning complication limiting it’s use. What about myocarditis and ileus? Both much deadlier than agranulocytosis.

- lines 250–251: what level of evidence does the use of clozapine have in the Polish guideline?

4.1. Mania

- Considering the partial D2/D3 agonistic properties of aripiprazole and cariprazine, and the higher risk of super-sensitivity psychosis with worsening of mania/psychosis should be mentioned. From my clinical experience, the use of cariprazine especially is potentially dangerous in patients with acute psychosis due to this specific property.

4.2.5. Aripiprazole

- line 335: “potentiation of antidepressive medication” is odd phrasing

5.

- line 377: “can be traced”, not “could”

- lines 385–388: D2 receptors are most likely just one part of the equation; most likely other mechanisms also play a significant role, especially considering quetiapine has a particularly low affinity for D2 receptors. Please rephrase so this becomes clear.

6.

- line 415: dopaminergic neurotransmitter isn’t increased in all areas of the brain, in some it is decrased (e.g., negative symptoms of schizophrenia). Therefore it may be more appropriate to write “altered” or “abnormal” 

Author Response

Responses to reviewer 1 – Brain Sciences

General comments:

-          Overall an interesting an informative manuscript that provides a compact overview of the evidence of antipsychotic drug use in various states of affective disorders. It is very well-structured and this easy to follow. I also enjoyed the historical aspects the author mentions.

Response: Thank you for the positive comments

-          The language in this manuscript would benefit from editing by a native speaker.

Response: The revised manuscript was checked by a native speaker, Mr. Richard Ashcroft

-          I appreciate the receptor pharmacological approach the author has taken and I personally believe that it is invaluable to understand how drugs work on the different receptor/receptor systems. However, I assume many readers are not as familiar with the receptors and their functions. Perhaps a brief table explaining their therapeutic effects when agonists or antagonists are used would be helpful.

Response: The association of antimanic and antidepressant activity with receptor effects was elaborated in detail in chapter 6.

-          The author of this manuscript has dedicated much work into the analysis of clozapine’s use in bipolar disorder. While I have heard that clozapine (like all other second generation antipsychotic drugs, SGA) has mood-stabilizing properties, I am not aware that this approach is considered routine clinical practice. It is also “off-label” as the summary of product characteristics reserves the use of clozapine for treatment-resistant schizophrenia and schizoaffective disorders. This should be mentioned as it can have serious legal implications!!

Response: Clozapine meets the criteria for mood stabilizer established by the author of this review (ref. 17). The use of the drug in treatment-resistant manic cases is suggested based on the recent reviews (ref. 36 and 370. In the Polish guidelines elaborated by the author of this review, clozapine is shown as a drug of the second choice in mania (ref. 39)

-          Several times I read “atypical” or “typical” antipsychotic drugs vs first-/second-generation. Please use these terms homogenously. Also consider introducing an abbreviation such as SGA for second-generation and FGA für first generation. It would improve readability.

Response: In the paper, a distinction of antipsychotic drugs into three generations is proposed. According to the reviewer’s suggestion, the abbreviations of FGA (first-generation antipsychotics), SGA (second-generation antipsychotics), as well as TGA (third-generation antipsychotics), are used.

  1. Introduction:

-          I do not understand what the first sentence means. Please rephrase to make it more comprehensible.

-          Please mind the use of correct tenses in the first paragraph: “As the date of the introduction of antipsychotic drugs to psychiatry, the publication of French psychiatrists led by Jean Delay in 1952 is assumed, showing the therapeutic (antipsychotic) effect of phenothiazine derivate, chlorpromazine [1]. An important year in the history of these drugs would be is also 1957 when the antipsychotic effect of haloperidol, the butyrophenone derivate, was demonstrated by Belgian psychiatrists [2]. Another significant date could be is 1963 when the publication of Swedish pharmacologists appeared suggesting that the pharmacological mechanism of chlorpromazine and 33 haloperidol is connected with their effect on the dopaminergic system of the brain [3].”

Response: This part of the introduction was changed according to the reviewer’s suggestions  

1.2. Second-generation of antipsychotic drugs

- I recommend starting this section with the development of the first SGA, i.e. clozapine

Response: With the advent of the SGA, the 1990s are assumed. Therefore I left the section as it was.

- “where it boomed a second childhood” is very odd phrasing

Response: it was replaced with “where it experienced a therapeutic revival”

- Line 90: Clozapine does not have a particularly high affinity for D2-receptors

Response: This is true but not relevant to the general idea.

- Line 95: Clozapine is definitely not a mood-stabilizing drug and its use as such is – at least currently– most certainly off-label! This sentence should be more clearly phrased, so it becomes more apparent, that clozapine was the first SGA that was found to have “mood-stabilizing” properties.

Response: Clozapine meets the criteria for mood stabilizer established by the author of this review (ref. 17).

- Line 95: perhaps better to say “chemically unrelated”

Response: I found it not necessary. E.g., olanzapine is chemically related to clozapine

- Lines 108–: what are the criteria for “mood stabilizers”?

Response: The criteria were established and published by the author of this article (ref.17).

- Lines 120–122: poor language, please rephrase

Response: The sentence was rephrased.

1.3. Third generation of antipsychotic drugs

- Lines 124–132: please provide appropriate references

Response: References 19-21 were provided

- Lines 134: instead of “are partial agonists of X” “act as partial agonists on XY”

Response: It was corrected according to the reviewer’s suggestion

- Lines 137–139: it would also be of value to mention that cariprazine has the highest affinity to D3-receptors

Response: It was mentioned.

- Lines 144: “Since their entered” is not correct English; perhaps you mean to say “since their introduction”?

Response: it was replaced with “Since their introduction”

2.1. Mania

- Lines 154–167: very interesting historical aspects!

Response: Thank you for appreciating it.

- Lines 169: Instead of calling haloperidol the “other most important” antipsychotic drug, it would be more objective to write that it was a “clinically valuable” antipsychotic drug.

Response: “most important” was replaced by “clinically valuable”

- line 175: Which other “typical” and “atypical” antipsychotics were considered in the mentioned meta-analysis?

Response: SGA – risperidone, olanzapine, quetiapine, asenapine, ziprasidone; TGA – aripiprazole (ref. 32)

- line 176–178: At least from where I am from, haloperidol is rarely used in the treatment of mania – this is not to say it isn’t effective. If haloperidol really does find significant use in mania today, please provide an appropriate reference.

Response: Reference 33 was provided.

 2.2. psychotic depression

- lines 180–181: “turned out” is informal English, please rephrase

Response: “turned out” was replaced with “emerged”

- line 183: “were used” implies that this was standard procedure in the past. When was this the case? 

Response: The sentence was rephrased.

3.1.1. Clozapine

- line 193–195: I would not go as far as to state that clozapine is “generally recommended” for the treatment of treatment-refractory mania, as this recommendation (to my knowledge) does not appear in the official treatment guidelines of other countries (NICE, etc.). Further, clozapine is solely in-label for treatment-resistant schizophrenia. In so, the use in bipolar patients is absolutely off-label, even though it may prove effective for this indication. 

Response: The fragment was changed

3.2.1. Clozapine

- lines 244–245: In most countries/according to the guidelines I am aware of, clozapine is not considered “second-line” treatment. The CANMAT guidelines you cited earlier consider clozapine “third-line” and clozapine has the lowest level of evidence for maintenance treatment of bipolar disorder. Further, clozapine’s risk for agranulocytosis is hardly the only concerning complication limiting its use. What about myocarditis and ileus? Both much deadlier than agranulocytosis.

Response: The fragment was rephrased.

- lines 250–251: what level of evidence does the use of clozapine have in the Polish guideline?

Response: The second level of evidence (ref. 39)

4.1. Mania

- Considering the partial D2/D3 agonistic properties of aripiprazole and cariprazine, and the higher risk of super-sensitivity psychosis with worsening of mania/psychosis should be mentioned. From my clinical experience, the use of cariprazine especially is potentially dangerous in patients with acute psychosis due to this specific property. 

Response: The reports were only provided on antimanic action of these drugs with better effect in combination with lithium or valproates (ref. 69-71)

4.2.5. Aripiprazole

- line 335: “potentiation of antidepressive medication” is odd phrasing

Response: this was replaced with “augmentation of antidepressants”

5.

- line 377: “can be traced”, not “could”

Response: The whole paragraph was changed

- lines 385–388: D2 receptors are most likely just one part of the equation; most likely other mechanisms also play a significant role, especially considering quetiapine has a particularly low affinity for D2 receptors. Please rephrase so this becomes clear. 

Response: The fragment was rephrased

  1.  

- line 415: dopaminergic neurotransmitter isn’t increased in all areas of the brain, in some it is decreased (e.g., negative symptoms of schizophrenia). Therefore it may be more appropriate to write “altered” or “abnormal” 

Response: “increased” was replaced by “altered”

Reviewer 2 Report

This submission reports a straightforward compendium of the role of antipsychotic drugs in the management of bipolar disorder. 

While it is very clear and useful for the reader, it somewhat overlooks more clinical, practical notions. Since the declared scope of this work is to talk about the “application” of antipsychotic drugs in mood disorders, I have a couple of suggestions that can hopefully increase the relevance of this paper in this sense.

1.     First, while each paragraph already provides some hints, I would properly synthesize the available evidence about direct comparisons between drugs. I would advise the Author to refer to Kishi et al., 2022 [https://doi.org/10.1038/s41380-021-01334-4] for mania, Bahji et al., 2020 [https://doi.org/10.1016/j.jad.2020.03.030] for bipolar depression, and Kishi et al., 2020 [https://doi.org/10.1038/s41380-020-00946-6] for maintenance treatment.

2.     Second, the differences between oral and long-acting formulations and their utility in clinical practice needs to be properly highlighted. The use of long-acting formulations of antipsychotics has been extended to people bipolar disorder in clinical practice, due to their usefulness in improving compliance, reducing the daily burden of oral formulations, and ultimately reducing the risk of recurrences [Keramatian et al., 2019, https://doi.org/10.1007/s40263-019-00629-z]. In this regard, real-world studies have demonstrated the efficacy of switching from orals to LAIs in the maintenance treatment of bipolar disorder. I would suggest that the Author refers to a recent, relevant mirror-image study highlighting how both first- and second-generation long-acting antipsychotics are effective in reducing hospitalization days and rates in bipolar disorder [Bartoli et al., 2022 https://doi.org/10.1007/s00406-022-01522-5].

Moreover, I was wondering if the manuscript may benefit from a reorganization, moving sections “5. Mechanism of therapeutic action of antipsychotic drugs in mood disorders” and “6. Pharmacological overlap between schizophrenia and bipolar disorder” before sections 3, 4, and 5.

Author Response

Responses to reviewer 2 – Brain Sciences

This submission reports a straightforward compendium of the role of antipsychotic drugs in the management of bipolar disorder. 

While it is very clear and useful for the reader, it somewhat overlooks more clinical, practical notions. Since the declared scope of this work is to talk about the “application” of antipsychotic drugs in mood disorders, I have a couple of suggestions that can hopefully increase the relevance of this paper in this sense.

  1. First, while each paragraph already provides some hints, I would properly synthesize the available evidence about direct comparisons between drugs. I would advise the Author to refer to Kishi et al., 2022 [https://doi.org/10.1038/s41380-021-01334-4] for mania, Bahji et al., 2020 [https://doi.org/10.1016/j.jad.2020.03.030] for bipolar depression, and Kishi et al., 2020 [https://doi.org/10.1038/s41380-020-00946-6] for maintenance treatment.

Response: The comparisons between SGA and TGA in mania, bipolar depression and maintenance treatment discussing the results of all three papers above (as ref. 82-84) were done in chapters 5.1, 5.2, and 5.3.

  1. Second, the differences between oral and long-acting formulations and their utility in clinical practice needs to be properly highlighted. The use of long-acting formulations of antipsychotics has been extended to people bipolar disorder in clinical practice, due to their usefulness in improving compliance, reducing the daily burden of oral formulations, and ultimately reducing the risk of recurrences [Keramatian et al., 2019, https://doi.org/10.1007/s40263-019-00629-z]. In this regard, real-world studies have demonstrated the efficacy of switching from orals to LAIs in the maintenance treatment of bipolar disorder. I would suggest that the Author refers to a recent, relevant mirror-image study highlighting how both first- and second-generation long-acting antipsychotics are effective in reducing hospitalization days and rates in bipolar disorder [Bartoli et al., 2022 https://doi.org/10.1007/s00406-022-01522-5].

Response: The use of LAI antipsychotics in the long-term treatment of bipolar disorder was discussed in chapter 5.4 (ref. 85-86).

Moreover, I was wondering if the manuscript may benefit from a reorganization, moving sections “5. Mechanism of therapeutic action of antipsychotic drugs in mood disorders” and “6. Pharmacological overlap between schizophrenia and bipolar disorder” before sections 3, 4, and 5.

Response: I decided to keep the present organization which I found more appropriate to my narration on this topic.

Round 2

Reviewer 2 Report

I cannot see the newly added references in the reference list. Otherwise I have no further comments.